# Accelerating Toeplitz Neural Network with Constant-time Inference Complexity

**Zhen Qin,   Yiran Zhong**[✉]

OpenNLPLab, Shanghai Artificial Intelligence Laboratory

`https://github.com/OpenNLPLab/ETSC-Exact-Toeplitz-to-SSM-Conversion`

## Abstract

Toeplitz Neural Networks (TNNs) have exhibited outstanding performance in various sequence modeling tasks. They outperform commonly used Transformer-based models while benefiting from log-linear space-time complexities. On the other hand, State Space Models (SSMs) achieve lower performance than TNNs in language modeling but offer the advantage of constant inference complexity. In this paper, we aim to combine the strengths of TNNs and SSMs by converting TNNs to SSMs during inference, thereby enabling TNNs to achieve the same constant inference complexities as SSMs. To accomplish this, we formulate the conversion process as an optimization problem and provide a closed-form solution. We demonstrate how to transform the target equation into a Vandermonde linear system problem, which can be efficiently solved using the Discrete Fourier Transform (DFT). Notably, our method requires no training and maintains numerical stability. It can be also applied to any LongConv-based model. To assess its effectiveness, we conduct extensive experiments on language modeling tasks across various settings. Additionally, we compare our method to other gradient-descent solutions, highlighting the superior numerical stability of our approach. The source code is available at https://github.com/OpenNLPLab/ETSC-Exact-Toeplitz-to-SSM-Conversion.

## 1 Introduction

Transformer has dominated the fields of computer vision (CV) (Dosovitskiy et al., 2020; Liu et al., 2021; Sun et al., 2022b), natural language processing (NLP) (Radford et al., 2018; Devlin et al., 2019; Radford et al., 2019; Brown et al., 2020; Liu et al., 2022), and speech processing (Karita et al., 2019;

---

[✉] Indicates the corresponding author (Email address: *zhongyiran@gmail.com*).

Zhang et al., 2020; Gulati et al., 2020; Sun et al., 2022a), becoming one of the best-performing approaches across different benchmarks. The core component of the Transformer, the attention mechanism, has a quadratic time complexity with respect to sequence length, making it challenging to scale to long sequences and large model sizes. Various methods have been proposed to address this issue, including Linear Attention (Katharopoulos et al., 2020; Choromanski et al., 2020; Qin et al., 2022b, 2023b), State Space Model (SSM) (Gu et al., 2022; Gupta et al., 2022), Toeplitz Neural Network (TNN) (Qin et al., 2023a) and other Long-Conv methods (Li et al., 2023).

Linear Attention reduces the space-time complexity of attention to linear by using a kernel trick to decompose the Softmax function (Choromanski et al., 2020; Qin et al., 2023c), but its poor performance (Qin et al., 2022a) prohibits it from being used to build Large Language Models (LLMs). SSM replaces the attention operation with state space equations, resulting in log-linear training space-time complexities (Gu et al., 2022). However, the performance of this method in casual language modeling is often inferior (Qin et al., 2023a) and initialization-sensitive (Gu et al., 2022), making it unsuitable for building LLMs.

TNN is a new class of sequence modeling methods that belongs to LongConv-based methods (Li et al., 2023; Qin et al., 2023a). It models long sequences using Toeplitz matrices to encode relative positional relationships. This key component allows them to effectively capture the dependencies within the sequence and make accurate predictions. It has a log-linear space-time complexity and outperforms Transformers in NLP and long sequence modeling tasks (Qin et al., 2023a). Additionally, its stable training capability and insensitivity to initialization make it feasible for LLMs.

Note that the above analysis has only taken into account the training complexities for the afore-

mentioned methods. However, when considering the deployment of LLMs, the inference complexities are also important. In decoder scenarios, *i.e.,* casual language modeling, the time complexity of inferring the $n^{\text{th}}$ token in the Transformer is $O(n^2d + nd^2)$, where $n, d$ are the sequence length and the feature dimension respectively. By using the KV cache technique (Pope et al., 2022), the complexity can be reduced to $O(nd^2)$. For Linear Attention, the complexity is $O(dh)(h$ is the hidden dimension), which makes it constant with respect to the sequence length (Katharopoulos et al., 2020). SSM also has a constant space-time complexity of $O(dh)$, where $h$ is the hidden space dimension (Gu et al., 2022). TNN, on the other hand, has a log-linear space-time complexity of $O(nd \log n)$ in inference, which may make it challenging to handle long sequences.

In this paper, we aim to accelerate the inference of TNN to constant-time complexity. We find that SSM can be thought of as a particular variation of TNN. TNN can benefit from the same inference complexity as SSM if we can convert it to SSM in inference. We show that such conversion can be viewed as an optimization problem and can be efficiently solved by a closed-form solution. Specifically, given a Toeplitz matrix, we first convert it to Vandermoode Linear System with Inclusive Equation Reformulation (IER) and then employ the Discrete Fourier Transform (DFT) to obtain a numerical stable result. Compared with gradient-based algorithms, our method is fast, training-free, and numerically stable. Note that our method can be applied to other LongConv-based methods (Li et al., 2023) as well.

We conduct extensive experiments to validate the effectiveness of our method. We compare our method with gradient-based methods in terms of efficiency and errors. Our method outperformed gradient-based methods significantly in efficiency while enjoying much lower error rates. We also apply our method to TNN language models and test it in real scenarios. Our method has equivalent extrapolation capabilities and perplexity to the original implementation of TNN. For the number of layers, sequence length, and feature dimensions, an in-depth assessment of speed and memory utilization is performed. Our method clearly outperforms the original TNN inference algorithm implementation. Furthermore, we demonstrate the applicability of our strategy beyond TNN by extending it to other LongConv-based models.

## 2 Background and Preliminary

In this section, we first define sequence model inference mathematically and then briefly discuss the inference complexities of Transformer and some closely related efficient sequence modeling methods such as Linear Transformer (Katharopoulos et al., 2020), SSM (Gu et al., 2022), and TNN (Qin et al., 2023a).

### 2.1 Inference

Inference refers to the process of predicting the next token given a language model $\mathcal{F}$ and a token sequence $\mathbf{x} \in \mathbb{R}^n$. It can be represented as follows:

$$\begin{aligned} \text{logits} &= \mathcal{F}(\mathbf{x}) \in \mathbb{R}^{n \times V} \\ x_{n+1} &= \text{Sample}(\text{logits}[-1]), \end{aligned} \tag{1}$$

where $V$ represents the size of the vocabulary, logits represents the output logits from the language model, and $x_{n+1}$ is the sampled token. The inference process continues until $x_{n+1}$ is the end-of-sequence token (eos), indicating the completion of inference. The time and space complexity of inference is determined by the underlying language model $\mathcal{F}$.

### 2.2 Inference Complexity

**Transformer** The Transformer's core component is self-attention, which operates on queries $\mathbf{Q}$, keys $\mathbf{K}$, and values $\mathbf{V}$. Each component is a linear mapping of the input $\mathbf{X} \in \mathbb{R}^{n \times d}$, given by:

$$\mathbf{Q} = \mathbf{X}\mathbf{W}_Q, \ \mathbf{K} = \mathbf{X}\mathbf{W}_K, \ \mathbf{V} = \mathbf{X}\mathbf{W}_V \in \mathbb{R}^{n \times d}. \tag{2}$$

The output of attention is computed as follows:

$$\mathbf{O} = \text{Softmax}\left(\frac{\mathbf{Q}\mathbf{K}^\top}{\sqrt{d}}\right)\mathbf{V}. \tag{3}$$

Due to the need to compute $\mathbf{Q}\mathbf{K}^\top$, the time complexity of Transformer is $O(n^2d + nd^2)$. During the inference phase, when predicting the $n$-th token, the naive time complexity is $O(n^2d + nd^2)$, with space complexity of $O(nd)$. By caching the previous time steps' $\mathbf{K}$ and $\mathbf{V}$, known as KV cache, the complexity can be reduced to $O(nd^2)$.

**Linear Transformer** The core component of the Linear Transformer is the Linear Attention, which uses the mapping $\phi(\cdot)$ to map the Query and Key to their implicit representations, where

$\phi(\mathbf{Q}), \phi(\mathbf{K}) \in \mathbb{R}^{n \times h}$ and $h$ is the hidden dimension. The output is then given by:

$$
\begin{aligned}
\mathbf{O} &= \mathbf{\Delta}^{-1}\phi(\mathbf{Q})[\phi(\mathbf{K})^\top \mathbf{V}], \\
\mathbf{\Delta} &= \operatorname{diag}(\phi(\mathbf{Q}))[\phi(\mathbf{K})^\top \mathbf{1}_n].
\end{aligned}
\tag{4}
$$

By first computing $\phi(\mathbf{K})^\top \mathbf{V}$, the computational complexity can be reduced to $O(ndh)$. During the inference phase, according to (Katharopoulos et al., 2020), we can transform the Linear Attention into the form of an RNN:

$$
\begin{aligned}
\mathbf{a}_0 &= 0, \mathbf{b}_0 = 0, \\
\mathbf{a}_n &= \mathbf{a}_{n-1} + \phi(\mathbf{k}_n)\mathbf{v}_n^\top, \\
\mathbf{b}_n &= \mathbf{b}_{n-1} + \phi(\mathbf{k}_n), \\
\mathbf{o}_n &= \frac{\phi(\mathbf{q}_n)^\top \mathbf{a}_n}{\phi(\mathbf{q}_n)^\top \mathbf{b}_n}.
\end{aligned}
\tag{5}
$$

This results in a time and space complexity of $O(hd)$ for the Linear Transformer.

**State Space Model** The State Space Model (SSM) (Gu et al., 2022) is to use state space equations for sequence modeling:

$$
\mathbf{u}_n = \mathbf{A}\mathbf{u}_{n-1} + \mathbf{B}x_n, y_n = \mathbf{C}\mathbf{u}_n
\tag{6}
$$

where:

$$
\begin{aligned}
\mathbf{A} &\in \mathbb{R}^{h \times h}, \mathbf{B} \in \mathbb{R}^{h \times 1}, \mathbf{C} \in \mathbb{R}^{1 \times h}, \\
x_n, y_n &\in \mathbb{R}, \mathbf{u}_n \in \mathbb{R}^{h \times 1}.
\end{aligned}
\tag{7}
$$

Here, $h$ represents the hidden dimension of the state space model. Note that we have swapped the positions of $x_i$ and $\mathbf{u}_i$ compared to (Gu et al., 2022) for notational consistency. By expanding the Eq. 6, we can write the SSM as:

$$
y_i = \sum_{j=0}^{i} \mathbf{C}\mathbf{A}^{i-j}\mathbf{B}x_j, i = 0, \dots, n-1.
\tag{8}
$$

This allows for parallel training and has a complexity of $O(nd \log n)$. SSM has demonstrated its effectiveness in many long sequence modeling tasks (Gu et al., 2022).

As a variance of SSM, DSS (Gupta et al., 2022) suggests that assuming $\mathbf{A}$ to be a diagonal matrix $\Lambda$ can mitigate the initialization sensitivity (Gu et al., 2022) while maintaining comparable model performance. In this case, the equation can be simplified as follows:

$$
\mathbf{C}\Lambda^i\mathbf{B} = \sum_{k=0}^{h-1} c_k b_k \lambda_k^i.
\tag{9}
$$

During the inference phase, due to the Eq. 6, the computational complexity is $O(hd)$.

**Toeplitz Neural Network and LongConv-based moethod** The Toeplitz Neural Network (TNN) introduces token mixing (Yu et al., 2021) using a relative positional matrix or Toeplitz matrix. The core computation can be expressed as follows:

$$
\mathbf{y} = \mathbf{T}\mathbf{x}, \quad \mathbf{x}, \mathbf{y} \in \mathbb{R}^n.
\tag{10}
$$

where:

$$
\mathbf{T} = \begin{bmatrix}
t_0 & t_{-1} & \cdots & t_{-n+1} \\
t_1 & t_0 & & \vdots \\
\vdots & & t_0 & t_{-1} \\
t_{n-1} & \cdots & t_1 & t_0
\end{bmatrix} \in \mathbb{R}^{n \times n}.
\tag{11}
$$

Using the Fast Fourier Transform (FFT), the matrix multiplication above can be computed in $O(nd \log n)$ time complexity, which makes the TNN's time complexity $O(nd \log n)$. During the inference phase, according to the Eq. 10, the complexity for predicting the $n^{\text{th}}$ token is $O(nd \log n)$. Since TNN can be viewed as a form of LongConv-based methods (Li et al., 2023), other LongConv-based methods have the same complexities.

# 3 Method

The inference of TNN exhibits a time complexity of $O(nd \log n)$ and space complexity $O(nd)$ for predicting the $n$-th token which poses challenges for scaling TNN to handle extremely long sequences in inference. In this section, we will present our approach to converting TNN into the form of SSM, aiming to improve generation speed and memory to a constant.

## 3.1 Problem formulation

In this section, we show the connection between TNN and SSM and formulate our problem mathematically. Considering a language modeling scenario, the token mixing process can be written as:

$$
y_i = \sum_{j=0}^{i} t_{i-j}x_j, \quad i = 0, \dots, n-1.
\tag{12}
$$

On the other hand, SSM can be represented as:

$$
y_i = \sum_{j=0}^{i} \mathbf{C}\mathbf{A}^{i-j}\mathbf{B}x_j, \quad i = 0, \dots, n-1.
\tag{13}
$$

Let $\bar{t}_i = \mathbf{C}\mathbf{A}^i\mathbf{B}$, the equation can be rewritten as:

$$
y_i = \sum_{j=0}^{i} \bar{t}_{i-j}x_j, \quad i = 0, \dots, n-1.
\tag{14}
$$

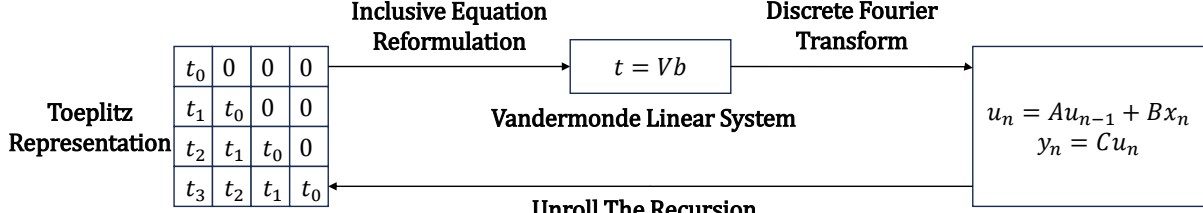

Figure 1: The conversion between Toeplitz representation and SSM representation. Unrolling the recursion can tranform SSM representation to Toeplitz representation. To obtain the inverse conversion, we use the Inclusive Equation Reformulation to express the problem as a Vandermonde Linear System. Then, we apply the Discrete Fourier Transform (DFT) to compute the SSM representation.

Since DSS is as effective as SSM (Gupta et al., 2022), but DSS has a simpler form, we choose DSS as our desired simplified structure. In this case, we have:

$$\bar{t}_i = \mathbf{C}\mathbf{A}^i\mathbf{B} = \sum_{k=0}^{h-1} c_k b_k \lambda_k^i. \tag{15}$$

Notably, $c_i b_i$ can be combined, so without loss of generality, we assume $\mathbf{C} = \mathbf{1}_h$:

$$\bar{t}_i = \mathbf{C}\mathbf{A}^i\mathbf{B} = \sum_{k=0}^{h-1} b_k \lambda_k^i. \tag{16}$$

By comparing the equations, it is evident that SSM is a special case of TNN. Since TNN inference encounters performance bottlenecks while SSM does not, the natural question arises: can we "convert" TNN to SSM in inference? This question is equivalent to find matrices $\Lambda$ and $\mathbf{B}$ such that:

$$t_i = \sum_{k=0}^{h-1} \lambda_k^i b_k, \quad i = 0, \ldots, n-1. \tag{17}$$

By determining suitable values for $\Lambda$ and $\mathbf{B}$, we can achieve an equivalent representation between TNN and SSM.

### 3.2 Gradient-based method

One solution to solve Eq. 17 is to use gradient-based methods to solve the following optimization problem:

$$\min_{b_k, \lambda_k} \sum_{i=0}^{n-1} \mathcal{L}\left(t_i, \sum_{k=0}^{h-1} \lambda_k^i b_k\right), \tag{18}$$

where $\mathcal{L}$ is the loss function, which can be $\ell_1$ or $\ell_2$.
However, this approach has two issues:
- It cannot exactly satisfy Eq. 17, resulting in information loss during the conversion.
- The presence of exponential terms $\lambda_k^i$ makes the optimization challenging to converge. (Gu et al., 2022)

The above issues make the gradient-based method less effective in achieving an accurate and efficient conversion from TNN to SSM. We adopt this algorithm as our baseline method and present it in Figure 2. The algorithm is summarized in Algorithm 2.

### 3.3 Our closed-form solution

In this section, we show that Eq. 17 can be directly solved with a closed-form solution, *i.e.,* find the exact values of $\lambda_k$ and $b_k$ that result in the desired Toeplitz matrix representation. With the closed-form solution, we can avoid the issues associated with the gradient-based approach and achieve a more accurate conversion from TNN to SSM.

To do this, we first add a variable $b = 0$ to both sides of the equation, yielding:

$$t_i = t_i + b = b + \sum_{k=0}^{h-1} \lambda_k^i b_k, \quad i = 0, \ldots, n-1. \tag{19}$$

Expanding this equation into matrix form, we have:

$$\begin{bmatrix} t_0 \\ t_1 \\ \vdots \\ t_{n-1} \end{bmatrix} = \begin{bmatrix} 1 & 1 & \ldots & 1 \\ 1 & \lambda_0 & \ldots & \lambda_{h-1} \\ 1 & \vdots & \ddots & \vdots \\ 1 & \lambda_0^{n-1} & \ldots & \lambda_{h-1}^{n-1} \end{bmatrix} \begin{bmatrix} b \\ b_0 \\ b_1 \\ \vdots \\ b_{h-1} \end{bmatrix}, \tag{20}$$

$$\mathbf{t} = \mathbf{V}\mathbf{b},$$
$$\mathbf{t} \in \mathbb{R}^n, \mathbf{V} \in \mathbb{R}^{n \times (h+1)}, \mathbf{b} \in \mathbb{R}^{(h+1)}.$$

Now, let's set $h = n - 1$, we have:

$$\begin{bmatrix} t_0 \\ t_1 \\ \vdots \\ t_{n-1} \end{bmatrix} = \begin{bmatrix} 1 & 1 & \ldots & 1 \\ 1 & \lambda_0 & \ldots & \lambda_{n-2} \\ 1 & \vdots & \ddots & \vdots \\ 1 & \lambda_0^{n-1} & \ldots & \lambda_{n-2}^{n-1} \end{bmatrix} \begin{bmatrix} b \\ b_0 \\ b_1 \\ \vdots \\ b_{n-2} \end{bmatrix}, \tag{21}$$

$$\mathbf{t} = \mathbf{V}\mathbf{b},$$
$$\mathbf{t} \in \mathbb{R}^n, \mathbf{V} \in \mathbb{R}^{n \times n}, \mathbf{b} \in \mathbb{R}^n.$$

At this point, $\mathbf{V}$ is a Vandermonde matrix. Although the Vandermonde linear system is un-

stable in general due to numerical precision issues (Gautschi, 2020), if the $\lambda_k$'s are pairwise distinct, the equation will have a solution. To improve stability, we can choose $\lambda_s = \exp\left(-\frac{2i\pi s}{n}\right)$, which results in $\mathbf{V} = \sqrt{n}\mathbf{W}_n$, where $\mathbf{W}_n$ is the Discrete Fourier Transform (DFT) matrix. The above equation can be expressed as:

$$\mathbf{t} = \sqrt{n}\mathbf{W}\mathbf{b}, \mathbf{W}^{\mathrm{H}}\mathbf{t} = \sqrt{n}\mathbf{b}, \quad (22)$$

where $\mathbf{W}^{\mathrm{H}}$ represents the conjugate transpose of the matrix $\mathbf{W}$. By comparing the first row, we have:

$$\sum_{i=0}^{n-1} t_i = 0. \quad (23)$$

However, the coefficients $t_i$ from TNN are not guaranteed to satisfy this equation. To ensure that this equation is satisfied, we introduce another variable $\bar{t}_n = -\sum_{i=0}^{n-1} t_i$, which we call an inclusive equation reformulation process. Therefore, we have:

$$\begin{bmatrix} t_0 \\ t_1 \\ \vdots \\ t_{n-1} \\ \bar{t}_n \end{bmatrix} = \begin{bmatrix} 1 & 1 & \cdots & 1 \\ 1 & \lambda_0 & \cdots & \lambda_{n-1} \\ 1 & \vdots & \ddots & \vdots \\ 1 & \lambda_0^n & \cdots & \lambda_{n-1}^n \end{bmatrix} \begin{bmatrix} b \\ b_0 \\ b_1 \\ \vdots \\ b_{n-2} \end{bmatrix}, \quad (24)$$

$$\mathbf{t} = \sqrt{n+1}\mathbf{W}_{n+1}\mathbf{b},$$

$$\mathbf{t} \in \mathbb{R}^{n+1}, \mathbf{V} \in \mathbb{R}^{(n+1)\times(n+1)}, \mathbf{b} \in \mathbb{R}^{n+1}.$$

Based on the above equation, we can determine the coefficients $b_i$ using the expression:

$$b_i = \frac{1}{\sqrt{n+1}}\left[\mathbf{W}_{n+1}^{\top}\mathbf{t}\right][i]. \quad (25)$$

By utilizing this formula, we can obtain the coefficients $b_i$. We name this method as ETSC (Exact Toeplitz-to-SSM Conversion) and provide a summary of the algorithm in Algorithm 1.

### 3.4 The inference of TNN

In this section, we briefly introduce three inference strategies of language modeling for TNN: the Original implementation, *i.e.,* FFT, Cache, and SSM. In the subsequent discussion, let us assume we have an L-layer TNN with the superscript $(l)$ indicating the result at the $l$-th layer. The computation of TNN can be represented as follows:

$$\mathbf{x}^0 = \mathrm{Embedding}(\mathbf{i}) \in \mathbb{R}^{n\times d},$$
$$\mathbf{x}^{l+1} = \mathbf{T}^l\mathbf{x}^l \in \mathbb{R}^{n\times d}, l = 0, \ldots, L-1 \quad (26)$$
$$\mathrm{Logits} = \mathbf{x}^L\mathbf{W} \in \mathbb{R}^{n\times V}$$

Here, $\mathbf{i} \in \mathbb{R}^n$ represents the input tokens and $V$ represents the vocabulary size.

---

**Algorithm 1** ETSC: Exact Toeplitz-to-SSM Conversion

**Input:** $\mathbf{t} \in \mathbb{R}^n$.
**Output:** $\lambda \in \mathbb{C}^n, \mathbf{b} \in \mathbb{C}^n$.
**Notation:** Use $\mathbf{W}_k$ to represent the $k$-th order DFT matrix.
**Initialize:**
$\bar{\mathbf{t}} = \mathrm{concat}([\mathbf{t}, -\sum_{i=0}^{n-1} t_i]) \in \mathbb{R}^{n+1}$,
$\lambda_s = \exp(-2\pi(s+1)/n + 1), s = 0, \ldots, n-1$,
$\bar{\mathbf{t}}_{\mathrm{dft}} = \mathbf{W}_{n+1}\bar{\mathbf{t}} \in \mathbb{R}^{n+1}$,
$\mathbf{b} = \mathbf{0}_n. \in \mathbb{R}^n$.
**for** $i$ in $0, \ldots, n-1$ **do:**
   $b_i = \bar{\mathbf{t}}_{\mathrm{dft}}[i+1]/\sqrt{n+1}$;
**end for**

---

**Algorithm 2** Gradient-Based Method

**Input:** $\mathbf{t} \in \mathbb{R}^n$;
**Output:** $\lambda \in \mathbb{C}^n, \mathbf{b} \in \mathbb{C}^n$;
**Initialize:**
$\mathbf{r}, \theta, \mathbf{b}_{\mathrm{real}}, \mathbf{b}_{\mathrm{img}}, \sim \mathcal{N}(0, \mathbf{I}_n)$.
**Minimize:**

$$\sum_i \left\| t_i - \sum_{k=0}^{h-1} \lambda_k^i b_k \right\|^2,$$

where

$$\lambda = \mathrm{Sigmoid}(r)\exp(i\theta),$$
$$\mathbf{b} = \mathbf{b}_{\mathrm{real}} + i\mathbf{b}_{\mathrm{img}}.$$

---

**Origin** In the inference phase, our core operation remains the computation of $\mathbf{T}^i\mathbf{x}^i$. One approach for inference is to continue using the Fast Fourier Transform (FFT), which results in a time complexity of $O(nd\log n)$.

**Cache** This method is to directly compute Eq. 12, which requires matrix multiplication and has a time complexity of $O(n^2d + nd^2)$. However, by employing a caching mechanism similar to the key-value (KV) cache in transformer (Pope et al., 2022), we can store the output of each layer as $\mathrm{cache}^l = \mathbf{x}^{l+1} \in \mathbb{R}^{n\times d}$. In this way, when performing a new inference, we only need to compute:

$$x_n^{l+1} = \sum_{k=0}^n t_{n-k}^{l+1} x_k^l. \quad (27)$$

Then, we update as follows:

$$\mathrm{cache}^l = \mathrm{concat}([\mathrm{cache}^l, x_n^{l+1}]),$$
$$x^{l+1} = \mathrm{cache}^l. \quad (28)$$

Table 1: **Extrapolation Evaluation on TNN.** We trained a TNN LM and, upon completion to training, utilized ETSC to convert the coefficients of the Toeplitz matrix into SSM representation. We then evaluated the model's extrapolation capability, comparing the results for different hidden states. It can be observed that our model exhibits extrapolation abilities similar to TNN. Moreover, for hidden states of 768 and 1024, ETSC achieves average perplexity (ppl) comparable to TNN.

| Dataset | Seqlen $h$ | 512 | 1024 | 2048 | 4096 | 8192 | 9216 | 10240 | 12288 | 14336 | AVG |
|---|---|---|---|---|---|---|---|---|---|---|---|
| wikitext-103 | TNN | 24.67 | 24.05 | 23.73 | 23.58 | 23.51 | 23.49 | 23.48 | 23.48 | 23.46 | 23.72 |
| | 512 | 24.65 | 24.47 | 24.37 | 24.32 | 24.29 | 24.29 | 24.28 | 24.28 | 24.28 | 24.36 |
| | 768 | 24.65 | 24.04 | 23.74 | 23.59 | 23.52 | 23.51 | 23.49 | 23.49 | 23.48 | 23.72 |
| | 1024 | 24.65 | 24.03 | 23.72 | 23.57 | 23.50 | 23.49 | 23.47 | 23.47 | 23.46 | 23.71 |
| wiki-book | TNN | 23.87 | 23.28 | 23.00 | 22.80 | 22.73 | 22.70 | 22.69 | 22.55 | 22.62 | 22.92 |
| | 512 | 23.87 | 23.28 | 23.00 | 22.80 | 22.73 | 22.70 | 22.69 | 22.55 | 22.62 | 22.91 |
| | 768 | 23.87 | 23.30 | 23.04 | 22.85 | 22.78 | 22.75 | 22.74 | 22.55 | 22.67 | 22.95 |
| | 1024 | 23.87 | 23.28 | 23.00 | 22.80 | 22.74 | 22.70 | 22.69 | 22.56 | 22.62 | 22.92 |

Table 2: **Evaluation of ETSC on Other LongConv Methods.** We conducted experiments to assess the performance of ETSC on other LongConv methods, specifically focusing on SGConv. We trained an SGConv LM and applied ETSC to convert the Toeplitz representation into SSM representation. We then evaluated the extrapolation capabilities of the converted model. This demonstrates that ETSC exhibits extrapolation abilities similar to SGConv, with even lower average perplexity (ppl) values.

| Seqlen | 512 | 1024 | 2048 | 4096 | 8192 | 9216 | 10240 | 12288 | 14336 | AVG |
|---|---|---|---|---|---|---|---|---|---|---|
| SGConv | 33.39 | 32.77 | 32.46 | 32.31 | 32.24 | 33.61 | 33.59 | 32.22 | 34.54 | 33.01 |
| Ours | 33.39 | 32.77 | 32.46 | 32.31 | 32.24 | 32.24 | 32.22 | 32.22 | 32.20 | 32.45 |

With this approach, the time complexity can be reduced to $O(nd^2)$.

**SSM** With our method, we can transform the Toeplitz representation into a State Space Model (SSM) representation. Therefore, we can perform inference using Eq. 6, resulting in both time and space complexities of $O(hd)$.

## 4 Experiments

In this section, we present extensive experiments to validate our method. We first analyze the numerical stability and efficiency of our method with a comparison to a gradient-based approach. Then we evaluate our method for language modeling tasks with real-world scenarios. In our inference efficiency study, we conduct an in-depth analysis of the impact of the number of layers, sequence length, and feature dimensions on the speed and memory utilization of our method. We also extend the scope of our method to other long convolution-based methods, showcasing its versatility and generalizability.

### 4.1 Numerical Stability and Efficiency

Figure 2 presents the comparison in terms of time complexity and relative error $\frac{\|\mathbf{t}-\mathbf{t}_{\text{pred}}\|}{\|\mathbf{t}\|}$, where $\mathbf{t} = [t_0, \ldots, t_{n-1}]$ represents the coefficients of the Toeplitz matrix. We first fix the feature dimension

to 64 and vary the sequence length from 64 to 8192. Our method is 3 to 6 orders of magnitude faster than the gradient-based method. Regarding the relative error, our method achieves an error close to zero, while the relative error of gradient-based methods exceeds 30%.

We then fix the sequence length to 2048 and vary the feature dimension from 64 to 16384. The gradient-based methods encounter OOM at $d = 512$ while our method successfully completes all tests. Our method is 4 orders of magnitude faster. In terms of relative error, our method achieves an error close to zero, while the relative error of gradient-based methods is around 35%.

Our method demonstrates superior numerical stability and efficiency compared to gradient-based methods. It significantly reduces the computation time while maintaining a low relative error. Furthermore, our method exhibits excellent scalability, as it can handle larger sequence lengths and higher feature dimensions without encountering OOM.

### 4.2 Evaluation on TNN LM

Following the configuration used in (Qin et al., 2023a), we trained a 6-layer TNN LM on the Wikitext-103 and Wiki-book (Wettig et al., 2023) dataset with a feature dimension of 512, maximum sequence length of 512, and 50k update steps. After training, we utilize ETSC to convert the coeffi-

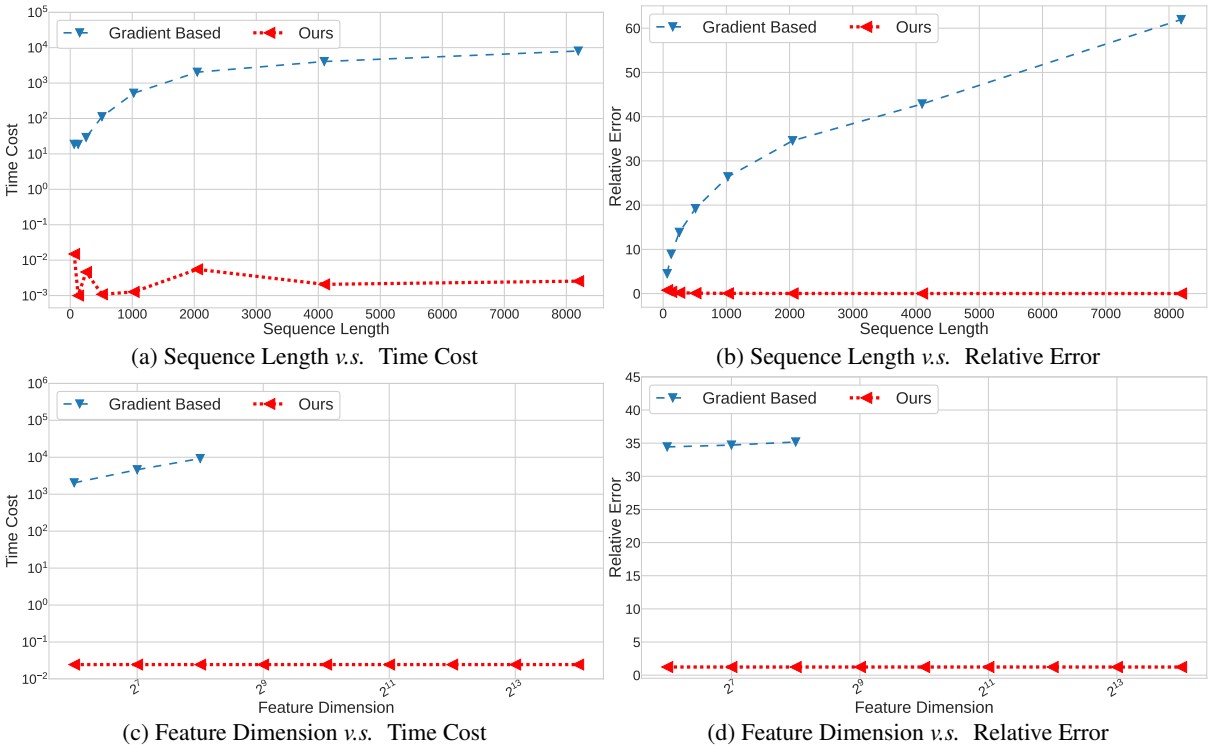

Figure 2: **Comparison of ETSC and Gradient-Based Methods.** We compare the time overhead and relative error $\frac{\|\mathbf{t}-\mathbf{t}_{\mathrm{pred}}\|}{\|\mathbf{t}\|}$ of ETSC and gradient-based methods, where the unit of time overhead is seconds and the unit of relative error is percent. Here, $\mathbf{t} = [t_0, \ldots, t_{n-1}]$ represents the coefficients of the Toeplitz matrix. It can be observed that ETSC exhibits significantly lower time overhead compared to gradient-based methods, while also achieving smaller errors.

cients of the Toeplitz matrix to SSM and vary the sequence length from 512 to 14336 to verify the model's extrapolation capabilities. We test with three hidden state dimensions: 512, 768, and 1024.

Table 1 shows the results of our evaluation. It can be observed that ETSC exhibits the same extrapolation capabilities as TNN, enabling it to handle sequences of arbitrary length. Moreover, when the hidden state dimensions are larger than 512, ETSC achieves comparable average perplexity to TNN, demonstrating ETSC preserves the modeling capacity of TNN while providing the benefits of numerical stability and efficiency.

Our evaluation on the TNN LM demonstrates that ETSC not only possesses extrapolation capabilities but also achieves comparable performance to TNN in terms of average perplexity. This further confirms the effectiveness and practicality of ETSC in long sequence modeling tasks.

### 4.3 Inference Efficiency Analysis

In this section, we discuss the impact of hyper-parameters on inference time and memory utilization. We compare ETSC with the Origin (FFT)

and Cache methods in terms of their practical inference time and memory usage. All methods are evaluated on the same A100 GPU. Specifically, we select a TNN LM and vary the sequence length, feature dimension, and number of layers to assess the effectiveness of the methods.

In the sequence length test, we fix the number of layers at 2 and the feature dimension at 64. In the feature dimension test, we fix the number of layers at 2 and the sequence length at 2048. In the layer test, we fix the sequence length at 2048 and the feature dimension at 64. Figure 3 (a) and (b) illustrate the results of the sequence length test. It can be observed that the Origin and Cache methods exhibit significantly higher inference times and memory utilization, ranging from several times to tens of times longer than ETSC. Additionally, the memory utilization of Origin and Cache is almost 2 orders of magnitude higher when the sequence length exceeds 1k. In the feature dimension test, as shown in Figure 3 (c) (d), both the Origin and Cache methods exhibit inference times several times to tens of times longer than ETSC, with memory utilization approximately 100 times higher. The layer test

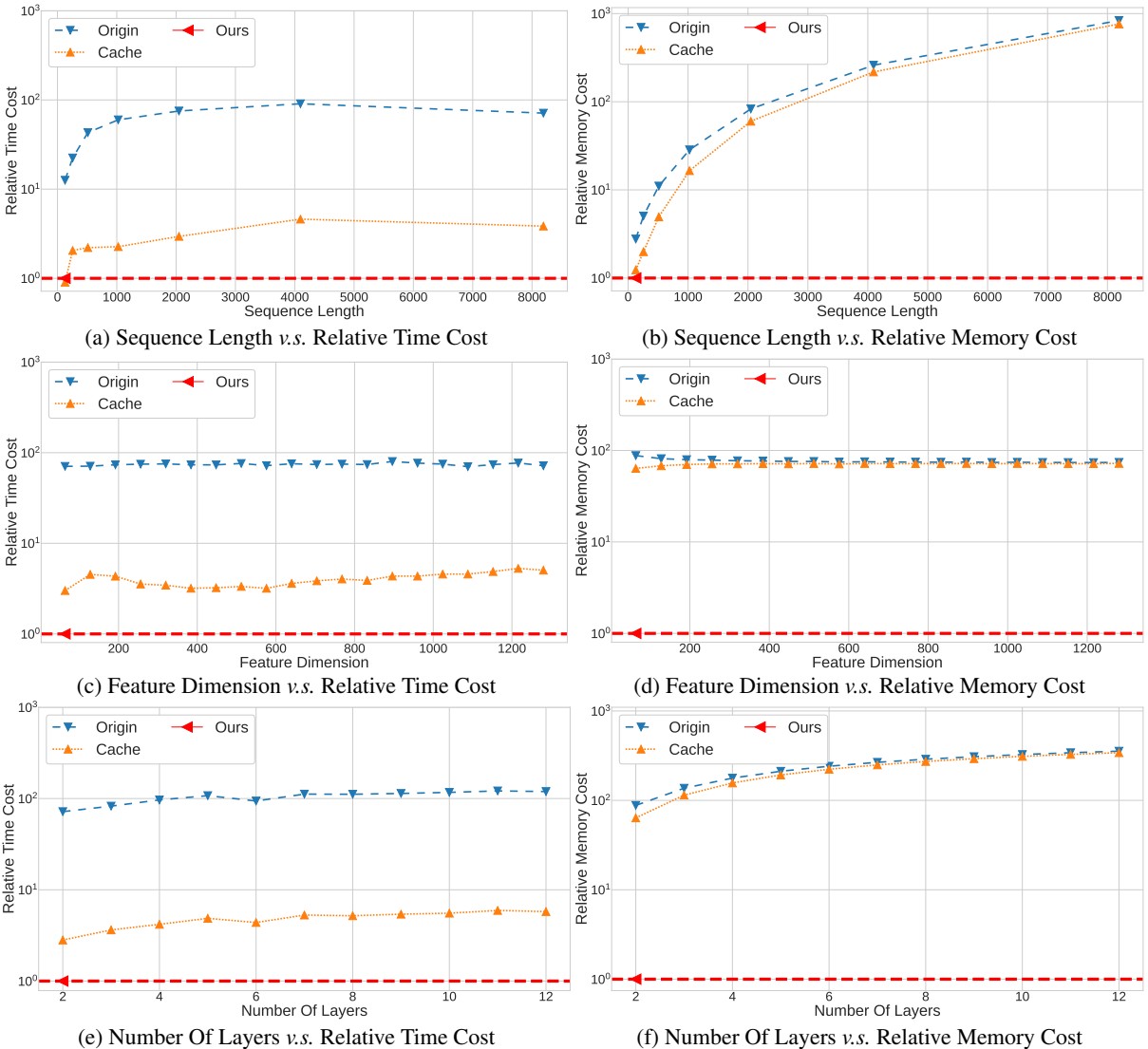

Figure 3: **Impact of Hyperparameters on Inference Time and Memory.** We compared the actual inference time and memory usage of ETSC, Origin (FFT), and Cache methods under different sequence lengths, feature dimensions, and model depths. Our method consistently outperformed the other methods, significantly reducing both the inference time and memory usage in all scenarios.

results are shown in Figure 3 (e) (f). The Origin and Cache methods again exhibit inference times several times to tens of times longer than ETSC, with memory utilization approximately 100 times higher or more.

These results demonstrate the superior efficiency of ETSC compared to the Origin and Cache methods across different configurations. ETSC consistently outperforms the other methods in terms of both inference time and memory utilization. This highlights the advantage of ETSC for efficient and scalable inference in long sequence modeling.

### 4.4 Application to Other LongConv-based Methods

Our method is applicable to all LongConv methods, as they all rely on Toeplitz matrices. To validate

this claim, we selected SGConv (Li et al., 2023) and trained an SGConv language model. After training, we used ETSC to convert the Toeplitz representation to the SSM representation. We then varied the sequence length in the range from 512 to 14336 to evaluate the model's extrapolation capabilities.

From Table 2, it can be observed that ETSC exhibits the same extrapolation capabilities as SGConv and achieves lower average perplexities. This indicates that our method can be effectively applied to other LongConv methods as well, further demonstrating its versatility and effectiveness in long sequence modeling tasks.

## 5 Conclusion

In this paper, we have analyzed and addressed the efficiency issue in TNN inference. We propose

a solution by converting the Toeplitz representation to the SSM representation, which reduces the time and space complexity of TNN inference to be independent of the sequence length. Our conversion algorithm, named ETSC, is fast, training-free, and numerically stable, outperforming other gradient-based methods significantly while keeping the same extrapolation capabilities and perplexity to the original TNN. We conducted a comprehensive assessment of the performance of our method in terms of the number of layers, sequence length, and feature dimensions. Our results clearly demonstrate that our method surpasses the original TNN in terms of both speed and memory utilization. Additionally, we extended the applicability of our strategy beyond TNN by successfully applying it to other LongConv-based models, showcasing the versatility and effectiveness of our approach.

## Acknowledgement

This work is partially supported by the National Key R&D Program of China (NO.2022ZD0160100).

## Limitations

While our proposed method for converting Toeplitz representations to State Space Models (SSM) has shown promising results in our experiments, there are certain limitations that should be acknowledged.

1. Trade-off between Accuracy and Efficiency: Although our method achieves significant improvements in efficiency, it is important to note that there may be a trade-off between accuracy and efficiency. The conversion from Toeplitz representations to SSM involves approximations and simplifications, which can introduce some level of error compared to the original representation. While our experiments have demonstrated comparable performance to the original Toeplitz Neural Network (TNN), there may be scenarios where the transformed SSM does not fully capture the intricate patterns present in the original model.

2. Application Scope: Our method has been extensively evaluated in language modeling tasks and demonstrated superior performance compared to gradient-based methods and the original TNN implementation. However, the applicability of our method may be limited to sequence modeling tasks and long convolution-based models. Further research is needed to explore its effectiveness in other domains and model architectures.

While our proposed method offers a compelling approach for converting Toeplitz representations to State Space Models, it is important to consider the limitations mentioned above. Addressing these limitations and further exploring the potential of our method in diverse domains and model architectures will be valuable directions for future research.

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
