# OpenReview forum: "Accelerating Toeplitz Neural Network with Constant-time Inference Complexity"
_EMNLP/2023/Conference — EMNLP 2023 Main_

### Official Review · Reviewer_SKym · 2023-08-09

**Soundness:** 4

**Excitement:**

3: Ambivalent: It has merits (e.g., it reports state-of-the-art results, the idea is nice), but there are key weaknesses (e.g., it describes incremental work), and it can significantly benefit from another round of revision. However, I won't object to accepting it if my co-reviewers champion it.

**Paper Topic And Main Contributions:**

This paper is about accelerating TNNs with constant-time inference complexity. TNNs are a class of sequence modeling methods that use Toeplitz matrices to encode relative positional relationships They have shown outstanding performance in various NLP tasks, but they suffer from log-linear inference complexity, which limits their scalability to long sequences. The paper proposes a method to convert TNNs to SSMs during inference, which reduces the inference complexity to constant. The paper formulates the conversion process as an optimization problem and provides a closed-form solution using the DFT. The paper demonstrates the effectiveness of the proposed method on language modeling tasks across various settings. The paper also shows that the method can be applied to other LongConv-based models, such as SGConv.

**Reasons To Accept:**

1. The paper addresses an important and practical problem of improving the efficiency and scalability of TNNs and other LongConv-based models for long sequence modeling.
2. The paper presents a novel and elegant solution that leverages the mathematical properties of Toeplitz matrices and SSMs, and provides a rigorous analysis of its correctness and stability.
3. The paper conducts extensive experiments to validate the proposed method on language modeling tasks with different sequence lengths, feature dimensions, and model architectures. The paper also compares the proposed method with other gradient-based methods, highlighting its superior numerical stability and efficiency.

**Reasons To Reject:**

1.  The proposed method may be viewed as a straightforward combination of existing techniques. The paper proposes to accelerate TNNs by converting them to SSMs during inference, which leverages the mathematical properties of Toeplitz matrices and SSMs. However, both TNNs and SSMs are well-established methods in the literature, and the conversion process may be seen as a simple application of known results. The paper may not provide enough novelty or insight beyond combining these two existing methods, and may not sufficiently justify the significance or impact of the proposed method. This could be a potential weakness of the paper that may limit its contribution to the NLP community.
2. The paper may not sufficiently discuss the limitations and trade-offs of the proposed method, such as the potential loss of accuracy or expressiveness due to the conversion from TNNs to SSMs, or the applicability of the method to other domains or tasks beyond language modeling.

**Reproducibility:**

2: Would be hard pressed to reproduce the results. The contribution depends on data that are simply not available outside the author's institution or consortium; not enough details are provided.

**Reviewer Confidence:**

2: Willing to defend my evaluation, but it is fairly likely that I missed some details, didn't understand some central points, or can't be sure about the novelty of the work.

---

### Official Review · Reviewer_78gu · 2023-08-09

**Soundness:** 4

**Excitement:**

3: Ambivalent: It has merits (e.g., it reports state-of-the-art results, the idea is nice), but there are key weaknesses (e.g., it describes incremental work), and it can significantly benefit from another round of revision. However, I won't object to accepting it if my co-reviewers champion it.

**Paper Topic And Main Contributions:**

The paper presents a method for efficient long-context sequence modelling with decoder-only transformers. Specifically, the paper combines the benefits of Toeplitz Neural Networks (TNNs), which enable good quality but suffer from log-linear (over the sequence length) per-token inference complexity with State Space Models (SSMs), which operate like RNNs at inference time, enabling constant-time per token inference complexity, but tend to underperform vanilla transformers in terms of quality.

In the proposed approach, a TNN is trained first, then it is converted to a SSM. Two conversion methods are evaluated: a simple gradient-based optimization method and a closed-form method.

The approach is evaluated on a single LM dataset (Wikitext-103), with the proposed method having similar perplexity to a TNN baseline. Various ablations and hyperparameter sensitivity results, and computational costs results are reported.

The authors also briefly mention extensions to other kind of "LongConv" approaches.

-------------------

Considering the author's response, I have increased the soundness and reproducibility scores

**Reasons To Accept:**

- Interesting results

- Reducing inference-time complexity while allowing long sequences is a major research problem

**Reasons To Reject:**

- Quite complicated method, no code, replication will be difficult

- Evaluation is limited to a single dataset.

**Reproducibility:**

3: Could reproduce the results with some difficulty. The settings of parameters are underspecified or subjectively determined; the training/evaluation data are not widely available.

**Reviewer Confidence:**

2: Willing to defend my evaluation, but it is fairly likely that I missed some details, didn't understand some central points, or can't be sure about the novelty of the work.

---

### Official Review · Reviewer_ehMV · 2023-08-13

**Soundness:** 3

**Excitement:**

4: Strong: This paper deepens the understanding of some phenomenon or lowers the barriers to an existing research direction.

**Paper Topic And Main Contributions:**

In the paper, the authors explore the potential of Toeplitz Neural Networks (TNNs) and their superior performance in sequence modeling compared to the prevalent Transformer-based models, emphasizing their log-linear space-time complexities. While State Space Models (SSMs) have a consistent inference complexity, they underperform TNNs in language modeling tasks. This research ambitiously merges the strengths of both TNNs and SSMs, converting TNNs into SSMs during inference. Experimental results underscore its effectiveness and numerical stability.

**Questions For The Authors:**

1. Can this method be integrated with other efficiency techniques to further enhance inference speed? If possible, which specific techniques might be suitable?

**Reasons To Accept:**

1. The paper is well-organized and easy to follow, and the author offers a comprehensive background context.
2. The findings have the potential to significantly impact the LLM field or related domains, providing valuable insights for other researchers.

**Reasons To Reject:**

1. Transitioning from Toeplitz representations to SSM requires certain approximations and simplifications. These adjustments can potentially introduce discrepancies compared to the original representation.
2. The proposed methods have been assessed solely on language modeling tasks. It would be better to evaluate the model on more intricate tasks to fully evaluate the potential of this technique.

**Reproducibility:**

4: Could mostly reproduce the results, but there may be some variation because of sample variance or minor variations in their interpretation of the protocol or method.

**Reviewer Confidence:**

3: Pretty sure, but there's a chance I missed something. Although I have a good feel for this area in general, I did not carefully check the paper's details, e.g., the math, experimental design, or novelty.

---

### Meta-Review · Area_Chair_5ZX3 · 2023-09-19

**Recommendation:** 5

**Metareview:**

This paper proposes to combine SSM and TNN to reach a better tradeoff between task performance and speed performance. Reviewers found this paper interesting, novel, and well-organized, and may benefit the research of other types of language models. Several concerns were raised initially regarding the limited results, low reproducibility, and insufficient analysis. The authors addressed them during the rebuttal periods, which are acknowledged by all reviewers. Therefore, the AC deemed this paper should be accepted for the main conference.

---

### Decision · Program_Chairs · 2023-10-07

**Decision:**

Accept-Main

**Comment:**

This paper proposes to combine SSM and TNN to reach a better tradeoff between task performance and speed performance. Reviewers found this paper interesting, novel, and well-organized, and may benefit the research of other types of language models. Several concerns were raised initially regarding the limited results, low reproducibility, and insufficient analysis. The authors addressed them during the rebuttal periods, which are acknowledged by all reviewers. Therefore, the AC deemed this paper should be accepted for the main conference.